# Nanoparticles (NPs) of WO_3-x_ Compounds by Polyol Route with Enhanced Photochromic Properties

**DOI:** 10.3390/nano9111555

**Published:** 2019-11-01

**Authors:** Marie Bourdin, Manuel Gaudon, François Weill, Mathieu Duttine, Marion Gayot, Younes Messaddeq, Thierry Cardinal

**Affiliations:** 1CNRS, Univ. Bordeaux, Bordeaux INP, ICMCB, UMR5026, 87 Avenue du Dr. Albert Schweitzer, 33608 F-Pessac CEDEX, France; marie.bourdin@u-bordeaux.fr (M.B.); Francois.Weill@icmcb.cnrs.fr (F.W.); mathieu.duttine@icmcb.cnrs.fr (M.D.); thierry.cardinal@icmcb.cnrs.fr (T.C.); 2Department of Physics, Center for Optics, Photonics and Lasers (COPL), Laval University, 2375 rue de la Terrasse, Québec, QC G1V-0A6, Canada; Younes.Messaddeq@copl.ulaval.ca; 3CNRS, Univ. Bordeaux, PLACAMAT UMS 3626, 87 Avenue du Dr. Albert Schweitzer, 33608 F-Pessac CEDEX, France; marion.gayot@u-bordeaux.fr

**Keywords:** polyol synthesis, nanoparticles, photochromism

## Abstract

Tungsten trioxide (WO_3_) is well-known as one of the most promising chromogenic compounds. It has a drastic change of coloration induced from different external stimuli and so its applications are developed as gas sensors, electrochromic panels or photochromic sensors. This paper focuses on the photochromic properties of nanoWO_3_, with tunable composition (with tunable oxygen sub-stoichiometry). Three reference samples with yellow, blue and black colors were prepared from polyol synthesis followed by post annealing under air, none post-annealing treatment, or a post-annealing under argon atmosphere. These three samples differ in terms of crystallographic structure (cubic system versus monoclinic system), oxygen vacancy concentration, electronic band diagram with occurrence of free or trapped electrons and their photochromic behavior. Constituting one main finding, it is shown that the photochromic behavior is highly dependent on the compound’s composition/color. Rapid and important change of coloration under UV (ultraviolet) irradiation was evidenced especially on the blue compound, i.e., the photochromic coloring efficiency of this compound in terms of contrast between bleached and colored phase, as the kinetic aspect is high. The photochromism is reversible in a few hours. This hence opens a new window for the use of tungsten oxide as smart photochromic compounds.

## 1. Introduction

Tungsten trioxide (WO_3_) is a well-known material studied for many years for its applications in gas sensing [1,2], electrochromism [3,4,5,6], photochromism [7,8,9,10], and photocatalysis [11,12]. It was recently shown [13] that oxygen sub-stoichiometry in tungsten oxide (with WO_3-x_ chemical composition) can have a strong impact on the optical properties on its optical properties. 

Tungsten trioxide has the same crystalline structure as rhenium oxide (ReO_3_) [14]. This structure can be described as a three-dimensional network of WO_6_ octahedra interconnected by their corners. Its various allotropic forms and sub-stoichiometric derivatives are known and listed in the literature. Indeed, the [WO_6_] octahedral site network can be slightly distorted in different ways with tungsten atoms located out of the center of the octahedron cage and/or twisted chains of octahedra. The various displacements of the cations (W^6+^ ions) and anions (O^2−^ anions) in the structure, and therefore the symmetry of WO_3_, can be modified with temperature [15]. Thus, depending on the temperature, the crystalline system of tungsten oxide can be tetragonal (P4/nmm), above 720 °C [16], orthorhombic (Pnmb) from 720 to 320 °C [17], monoclinic (P2_1_/n) from 320 to 17 °C or triclinic (P-1) below 17 °C [15]. Hexagonal and cubic structures are also listed in the literature. The hexagonal WO_3_ can be obtained by dehydration at 420 °C of hydrated tungsten trioxide (WO_3_.nH_2_O) [18]. The formation of cubic WO_3_ was observed at high temperature during the dehydration of aluminum tungsten phosphate (AlPW_12_O_20_) [19]. It is also known that tungsten oxide forms a set of sub-stoichiometric oxides with a general composition of W_n_O_3n-x_ with x = 1, 2 or 3 [20]. Oxygen sub-stoichiometry is known to potentially induce the occurrence of shear planes in the octahedron framework, leading to octahedra not only connected by their corners but also by their edges [21]. 

Regardless, the mainly described phase is still the monoclinic one, which is thermodynamically stable at room pressure and temperature. This structure exhibits some octahedral site twists that are different depending on the unit-cell axis. The different chevron patterns constituted by the octahedral framework are shown from the different *a*, *b* and *c* directions in Figure 1i–iii. The angle of the chevrons being different along the crystallographic axes, the unit-cell parameters of a, b and c are then all different: 7.32 Å; 7.56 Å and 7.73 Å, respectively, according to the crystallographic data of the literature [22]. It can be noted that the distinction between a monoclinic or an average cubic unit cell (the equivalent pseudo-cubic unit-cell is represented in Figure 1iv) may be difficult to discern for nano-sized WO_3_ particles, leading to an intrinsic enlargement of the X-ray diffraction peaks, often associated with surface defects as well as oxygen sub-stoichiometry.

Different synthesis routes are used for the preparation of WO_3_ oxides in the literature. Different approaches can be conducted for the direct obtaining of thin or thick films on a substrate, such as pulsed laser deposition [14], chemical vapor deposition [23] or spray pyrolysis [24]. Synthesis processes allows for the development of powders and suspensions by precipitation [25], hydrothermal [26] or polyol routes [27]. The different “soft chemistry” synthesis methods allows for the obtainment of particles with a large variety of morphologies such as nanorods [28], nanosheets [26] or even nanoflowers [29] and with sizes from 10 [30] to 200 nm [25].

The present article focuses on nanoparticles of WO_3_ obtained by polyol route. The polyol method is a robust strategy for the preparation of well-defined metal or oxide nanoparticles (NPs) in terms of size, shape, composition, and crystallinity. Typical synthesis entails the reduction of a metal precursor by polyol in the presence of an appropriate capping agent at an elevated temperature. In 1988, Figlarz et al. [31] developed the polyol method to synthesize metal powders, including Co, Ni, and Cu particles. The choice of the polyol type is dictated by the optimum reflux temperature and its reducing strength allowing the precipitation of well crystallized particles of metals oxide or metal particles. Hence, an optimal use of polyol process, thanks to the control of the reducing capacity of the polyol solvents, can give access to oxides having an oxygen sub-stoichiometry, i.e., favoring the lowest oxidation numbers without achieving the full reduction of the metallic elements [27]. 

The as-prepared particles were characterized by X-ray diffraction (XRD), transmission electron microscopy (TEM), electron paramagnetic resonance (EPR) and diffuse reflectance spectroscopy. The particles were then heat treated at different temperatures and under different oxygen partial pressures to show the influence of these treatments on their chemical composition (control of the x sub-stoichiometric coefficient in WO_3-x_ oxides), the obtained crystallographic structure, their morphology (particle size and shape) and their optical properties. Achievement of NPs with enhanced photochromic properties is finally discussed in regard of the recent progresses on the photochromic behavior of WO_3_ NPs in literature. The enhanced photochromic activity of the WO_3_ NPs obtained by polyol shown in this study represents a promising step towards scaled-up production of WO_3_-based thin films for smart windows or UV-sensing application. 

## 2. Materials and Methods 

### 2.1. Polyol Synthesis of WO_3_ Powders

Chemical reagents were purchased from Sigma Aldrich and used as received. Tungsten (VI) chloride was used as tungsten source and diethylene glycol (DEG) as solvent. WCl_6_ (7,2 g) was added to 100 mL of DEG and 20 mL of distilled water. The mixture was heated at 180 °C with continuous stirring and refluxed for 3 h. At the end of the reaction, a deep blue precipitate was obtained. The precipitate was washed and centrifuged several times with ethanol to remove any trace of solvent and dried in an oven at 80 °C. In order to study the influence of the heat treatment parameters, the as-synthesized powders were annealed at 100, 200, 300, 400 and 600 °C under air atmosphere or annealed at 600 °C under oxygen partial pressure from 10^−2^ bar to 10^−5^ bar (thanks to a calibrated mixtures of argon flow with 10^−5^ bar and oxygen partial pressure and gas flow). The three extreme samples: i.e., the raw powder, and the two compounds obtained after post-annealing treatments at 600 °C under air and argon atmospheres were chosen as main references for discussion.

### 2.2. Characterization Techniques

WO_3-x_ chemical composition of the as-prepared powders can be determined thanks to chemical titration and EPR measurements.

The chemical titration was made after dissolution of the powder in NaOH aqueous solution within the presence of KMnO_4_ and KI salt dissolved in large excess in regard to the tungsten trioxide concentration. A redox process between MnO_4_^−^ and W^5+^ ions and then between the rest of MnO_4_^−^ and I^−^ lead to the formation of I_2_ species, which can be titrated by Na_2_S_2_O_3_ in order to get W^5+^ concentration, and then to deduce the oxygen sub-stoichiometry in WO_3-x_ compounds (x equals half time the W^5+^ concentration). All reagents were provided by Merck, Darmstadt, GERMANY. 

Electron paramagnetic resonance experiments were performed from room temperature down to T = 5 K in order to identify the W5+ ions and/or the occurrence of free electrons in the conduction band of the tungsten trioxide semi-conductors. The presented EPR spectra were recorded with a Bruker EMX spectrometer operating at X-band frequency (9.45 GHz) with 1 mW microwave power, 0.5 mT magnetic field modulation amplitude (frequency 100 kHz) and a spectral resolution of 0.15 mT/pt.

The compound structures were characterized by X-ray diffraction analysis (PANanalytical X’Pert Pro instrument Cu K_α1_ = 1.54056 Å, K_α2_ = 1.54439 Å and 2θ range from 8 to 80°, Almelo, NETHERLANDS). The unit cell parameters were refined by structural pattern matching using the Fullprof program package. All the patterns were analyzed using the Caglioti function, i.e., the isotropic peak profile function for which the u, v, w and shape parameters are refined.

The morphology of the particles was studied using transmission electron microscopy (TEM JEOL 2100 and JEOL 1400+ microscopes, Tokyo, JAPAN) working at 200 or 120 kV acceleration voltage, respectively. HRTEM measurements were performed on the JEOL 2200FS microscope. Samples were prepared by dispersing a few milligrams of powder in ethanol with ultra-sonication and by directly depositing one drop of the as-prepared suspension on a carbon coated grid.

Morphological characterizations are also performed thanks to BET measurements with the Micromeritics Smart VacPrep for degassing and the Micromeritics Tristar II PLUS for the dinitrogen physisorption (Micromeritics, Merignac, FRANCE).

Diffuse absorption spectra were recorded at room temperature from 200 to 2000 nm on a Cary 5000 spectrophotometer (Agilent, Santa-Clara, CA, USA) using an integration sphere (spectral resolution: 1 nm and band length: 2 nm). Halon was used as the white reference. RGB space colorimetric parameters were determined from the spectra using a two-step mathematic treatment. The first step consists in extracting the XYZ tri-stimulus values (defined by the CIE, 1964) from the integration (over the visible range, i.e., from λ = 380 up to 780 nm) of the product of x(λ), y(λ) or z(λ) functions (CIE–1964) with the diffuse reflectance spectra function X = ∫ x(λ).R(λ)dλ. Then, we used the transfer equations defined by the CIE, 1976, to transform the XYZ space to the L*, a* and b* common three-color space parameters.

## 3. Results

### 3.1. Raw Powder and Post Annealed Samples under Argon and Air (600 °C)

The comparison of three reference samples is first discussed: the raw powder directly issued from the polyol synthesis; “blue powder”, the sample obtained from a post-annealing treatment under air at 600 °C; “yellow powder”, of the as-prepared raw powder, and the sample resulting from a post-annealing treatment under argon (with pO_2_ ≅ 10^−5^ atm) at 600 °C; “black powder”.

The structure of the three powders was characterized by X-ray diffraction. The XRD diagrams show that the raw powder and the argon-annealed powder have crystallized with a cubic Pm-3m structure and the powder annealed under air with a P2_1_/n monoclinic structure (Figure 2a). A refinement of the raw powder and the air-annealed powder was performed in order to determine the lattice parameters (Figure 2b–d).

The pattern matching of the air annealed sample (yellow sample) can be achieved without any difficulties using the P2_1_/n space group. The matching quality is good, and the reliability factors below 10%: Rexp = 3.01%; Rf = 8.55%; Rp = 6.47%. In opposition, the refinement of the raw powder firstly performed considering that a single cubic structure leads to a low matching quality with relatively high reliability factors: Rexp = 1.49%; Rf = 17%; Rp = 13.5%. The subtraction of the simulated signal to the experimental indicates that a “back-asymmetry” of the experimental diffraction peaks is observed, which cannot originate from the condition of the X-ray pattern acquisition (in opposition with “front-asymmetry”, which typically originates from the approximation made during the projection of the intercept of diffraction cones on the 2θ diffraction angle linear axis). Back-asymmetry cannot be issued from geometric consideration on the experimental measurement collection (on the contrary to front-asymmetry, which is caused by the 2D projection of the arc of the diffraction cone in Bragg-Bentano geometry). To take into account this back-asymmetry, one has to consider a distribution of the unit-cell parameters, i.e., to fit the experimental diffractogramms, a set of different unit-cell must be integrated to the calculation. For illustration, three lattice parameters instead of a single one are used herein. The result of this latter fitting is closer to the XRD experimental pattern, leading to a significant decrease of the various reliability factors: Rexp = 1.49%; Rf = 8.72%; Rp = 6.59%. The diffraction peaks width of the blue and the black samples is characteristic of low dimension crystallites. Thus, the raw powder seems to have a lattice parameter gradient. This can be explained considering two different hypothesis:

(i) Each nanometric particle constituting the blue and black samples exhibits a gradient of unit-cell parameters. It could be reasonnably considered that this unit-cell parameter gradient is created from surface proximity. Indeed, it was demonstrated in a previous work on ZnO nanoparticles synthetized by polyol route that the crystallites exhibit a unit-cell gradient from the surface to the bulk [32].

(ii) The different crystallites do not exhibit the same unit-cell paramters. The difference in unit-cell parameters from one to another may be explained by slight differences in terms of chemical composition; for instance, different oxygen sub-stoichiomety.

TEM micrographs of the different samples are reported on Figure 3. The blue sample particles (Figure 3c) are very small with a size around 5 nm. The particle size distribution width is quite narrow (between 4 and 7 nm), but the agglomeration of the particles and their superimposition tend to limit the accuracy of the particle size distribution calculation. This very small size is in good agreement with the previous discussion extracted from the X-ray diffraction patterns. Figure 2b shows that for the black samples, two morphologies can be well distinguished: isotropic nanoparticles with a diameter in the range 5–10 nm and anisotropic particles (rod-like particles) with a diameter about 10 nm and a length that can reach about 100 nm. Finally, the air-annealed powder (yellow sample) shows significantly larger isotropic particles of about 50 nm average diameter.

BET measurements were performed on the three samples. Sw surfaces extracted from BET are 12.25, 64.55 and 155.45 m^2^/g for the yellow, the black and the blue samples, respectively. Calculation of the crystallite size was performed considering in first approximation that (i) the whole surface of the crystallite is accessible, (ii) all the crystallites have a spherical shape and (iii) all the crystallites have a same diameter. The crystallite diameter is deduced from the expression: Sw = 6 × (WO_3_)/D, with Sw the specific area of the compound, (WO_3_) the oxide density and D the crystallite diameter. The crystallites diameters D calculated for the yellow, the black and the blue samples are equal to 68.4, 13 and 5.4 nm, respectively. This result is in perfect agreement with the TEM photographs.

The occurrence of some rod-like particles in the black samples is in fact due to the high reactivity of the particles while exposed to the electron beam during the TEM observation. Indeed, the focalisation of the electron beam for a few seconds to a few minutes on agglomerates of primary particles clearly leads to a very sudden and unexpected dissociation of the agglomerates and the growth of numerous rod-like crystallites, which are scattered around the initial agglomerate and then partially destroyed (Figure 4).

HRTEM investigations performed on the obtained rod-likes particles show that they are mono-crystalline, with a lattice spacing perpendicular to the rod long-axis in the range 0.335–0349 nm (Figure 5). This is not consistent with the d-spacing of the (100) plane of the cubic structure observed in X-ray diffraction patterns about 3.7–3.8 Å. Regardless, an anisotropic crystal shape is also not coherent with a cubic unit-cell. It can be observed that the WO_3-x_ compound is reported with numerous crystallographic unit-cells in literature. Indeed, numerous phases with WO_3-x_ general composition with x changes from 0.08 up to 0.375, have been experimentally observed. They are W_32_O_84_, W_3_O_8_, W_18_O_49_, W_17_O_47_, W_5_O_14_, W_20_O_58_, W_25_O_73_, and are known as Magneli phases [33]. These phases are characterized by the periodic organization of the oxygen vacancies associated with the periodic formation of [WO5] pentagonal bipyramids and/or [WO6] edge-sharing octahedrons with typically crystallite shapes such as nanowires, nanorods or nanosheets. Most of these phases have a large unit cell in which there are interplanar distances in the range 0.335–0.349 nm. Otherwise, hexagonal tungsten oxide nanorods, obtained by spray pyrolysis, were described by Nakakura et al. [34]; the hexagonal phase is based on (100) d-spacing equal to 0.67 nm, two times the herein observed lattice spacing, which is not so far. In conclusion, the rod-like particles formed under electron beam focalisation surely originate from a photo-electron reduction associated with a drastic phase transformation; the mostly sub-stoichiometric oxide (black sample prepared under the most reducing conditions) is logically expected to be the most reactive.

EPR was chosen to demonstrate the difference in tungsten–oxygen stoichiometry between the yellow, the blue and the black samples (Figure 6). At room temperature, the EPR spectrum of the sample annealed under argon (black) exhibits a narrow (peak-to-peak linewidth 0.6 mT) and intense symmetrical resonance line with Lorentzian shape. This isotropic EPR signal with g-factor, g = 2.0026 ± 0.0002, close to the value of the free electron (g_e_ = 2.0023) may be associated with conduction electrons rather than paramagnetic centres (such as electron trapped at oxygen vacancy) which normally exhibit anisotropic signals (axial or orthorhombic). At a low temperature (5 K), additional weak orthorhombic signals are detected at higher magnetic fields (350–420 mT region). These last EPR signals characterized by g-values close to 1.7 may be due to W^5+^ (d^1^) ions in distorted environments, most likely located close to the crystallite surface [35,36]. For the raw (blue) sample, the axial EPR signal with *g*_⊥_ = 1.76 and *g*_//_ = 1.71 is observed at low temperature (5 K) and can unambiguously be associated with W^5+^ ions in a more regular site (most likely with C_4v_ symmetry) within the WO_3_ structure [37,38,39].

The polyol synthesis, being carried out in a reducing medium in the presence of W^5+^ ion in the raw powder can be easily explained, as the germination–growth steps in diethylene glycol in reducing alcohol resulted from the oxy-condensation of both W^6+^ ions (mainly) and W^5+^ ions (in a minor way). The particles obtained are therefore not stoichiometric WO_3.0_, but clearly exhibit an oxygen sub-stoichiometry with a chemical formulae WO_3-δ_. Interestingly, the EPR signal does not show the occurrence of free electrons; it can be considered that all the oxygen vacancies are associated with local O^2−^ anions, which are associated to each other with two local W^5+^ cations, to form a “triple defect” forming two W^5+^ consecutive ions linked by an oxygen vacancy bridge. On the contrary, the post-annealing treatment under argon (about 10^−5^ oxygen partial pressure) can produce (i) an oxygen exfoliation at the surface of the NPs, leading to an additional reduction associated to a segregation of tungsten W^5+^ at the surface of the material, or (ii) the migration of the oxygen deficiencies from the bulk to the surface of the materials. In the black sample, the oxygen surface exfoliation leads to supernumerary electrons in regard to the oxygen vacancies in the crystallite core, hence allowing the occurrence of free electrons (oxygen vacancies being known to act as electron traps). A schematic representation of the blue and the black samples, with simplified crystallographic representation and associated energy band diagrams is proposed (Figure 7).

Finally, no EPR signal was detected for the powder annealed under air (yellow sample). This confirms that the monoclinic phase obtained after high temperature post-treatment under air exhibits a chemical composition very close to the perfect stoichiometric tungsten trioxide WO_3.0_. A post-annealing treatment under air produces the complete filling of the oxygen deficiencies leading to the stoichiometric WO_3_ product (yellow sample).

In order to determine the amount of W^5+^ within the raw material, an iodometric titration was performed on the blue sample powder, which can easily be dissolved in aqueous solution. The blue sample was dissolved with a soda pellet and an excess of potassium iodide. The I_2_ reaction product formed from the redox process between I^−^ and W^5+^ ions is then titrated with sodium thiosulfate (S_2_O_4_^2−^ reducing ion). The tests gave an average delta value of 0.018, leading to the formula of the raw powder W^6+^_0.982_W^5+^_0.036_O_2.982_.

Finally, diffuse reflectance spectroscopy allows us to determine the optical properties and the colorimetric coordinates of the powders in the CIE L*a*b* space (Figure 8). The yellow powder has a very slight absorption rate in the near infrared part of the spectrum and a very strong UV absorption; the reflectance spectrum versus wavelength can in first the approximation be described as an abrupt Boltzmann curve, characteristic of a wide-gap semiconductor, with the gap being located at 380 nm. The slight absorption in the infrared part of the spectrum, with an absorption coefficient continuously increasing with the increase of the source radiation wavelength (decrease of the energy) can be attributed to a very low density of free carrier charges thermally activated (from Valence Band to Conduction Band) which can be described as a Drude electronic cloud [40,41]. This yellow sample has a quasi-total reflection window in the visible spectrum, up to nearly 90% reflection at 500 nm; the yellow color being explained by the occurrence of the tungsten oxide gap located on the frontier between the UV range and the visible range, inducing a slight absorption of the purple hues. The argon-annealed powder (black sample) absorbs almost all of the light over the entire UV-Visible-NIR wavelength range (200–2000 nm). It was previously discussed how the oxygen sub-stoichiometry is then associated with the creation of free electrons causing the absorption of all the photons of the radiation energy range. The powder obtained directly after precipitation (blue sample) has interesting intermediate properties: it shows a strong absorption of the signal in both UV and the near infrared ranges, whereas a small window of reflectance in the visible range can be observed with a maximum at around 450 nm. The difference in the color of the products is due to the presence or absence of free electrons, but only from the intervalence band between localized W^5+^ ions and neighboring W^6+^ ions inside the bulk structure. Indeed, a small amount of W^5+^ is enough to produce a strong coloration of the product from intervalence mechanism (polaron transfer between consecutive W^5+^ and W^6+^ ions through oxybridges).

To control the sample reflectance in the visible range while also maintaining a strong absorption in the ultraviolet and near infra-red ranges is an interesting challenge for opening various applications (as solar filter, for instance). It is possible by modulating the sub-stoichiometry amount in oxygen, which is responsible for the strong variations of the optical properties observed. For such a proposal, two solutions have been considered: the first one consists in varying the post-annealing temperatures in air atmosphere to limit the re-oxidation of the raw sub-stoichiometric sample (blue sample); the second one is to perform post-annealing treatment at a fixed temperature of 600 °C while the redox conditions are balanced, thanks to the control of the annealing treatment atmosphere (with different oxygen partial pressures).

### 3.2. Effect of the Post Annealing Temperature under Air

After synthesis, the particles were washed and dried in an oven (80 °C), and then heat-treated in air at temperatures of 100, 200, 300, 400 and 600 °C. The crystalline structure of the powders has been determined by X-ray diffraction and the patterns obtained are shown in Figure 9.

The patterns show that, for powders annealed between 100 and 300 °C, the crystalline structure remains the same as the raw blue powder (with very low crystallinity, which corresponds to cubic-type phase) (Figure 2). The crystal structure changes from Pm-3m pseudo-cubic group to P2_1_/n monoclinic group when a post-annealing treatment over 400 °C is applied. However, the particles after annealing at 400 °C are significantly less crystallized than those annealed at 600 °C, since the peak widths of the pattern of the compound obtained at a low temperature are about two times larger than those obtained after annealing at 600 °C.

The diffuse reflectance spectra and the associated chromatic coordinates in L*, a*, b* space of the powders annealing at different temperatures under air are reported in Figure 10. There is no optical difference between the raw product and the product annealed at 100 °C—the two spectra are almost superimposed, and the colorimetric coordinates are similar. In regard to these two references and the first samples, the spectra of the powder obtained at 200 °C and 300 °C show: (i) a new absorbing contribution in the visible range, around 600 nm, which may be associated with the presence of carbon radicals that were revealed by EPR measurement, (ii) a large increase of the reflectance intensity (decrease of the large absorption band) in the NIR part of the spectrum corresponding to a decrease of the W^5+^ ions concentration due to the oxidative conditions of the post-annealing treatment, as observed by EPR. Carbon traces, despite the washing steps of the raw sample, remain in the raw powder. The very viscous di-ethylene-glycol used as solvent, with a furthermore high ebullition point (245 °C), is in fact very difficult to fully eliminate. Additionally, the positive capillary effect produced by the intra-agglomerate porosity, which can promote the agglomeration of the WO_3_ NPs particles and was observed by TEM, contributes to limit the removal of the polyol traces in the raw samples. Hence, during the heat treatment, organic compounds, most likely from the polyol solvent, have been partially degraded and radical carbons residues can be detected in the material. This signal is not detected for the powders annealed at 100 °C because the temperature is not high enough to initiate the degradation of the polyol (di-ethylene glycol has a boiling point of 245 °C at room pressure under air). The product annealed at 400 °C shows a diffuse reflection signal very similar to the 600 °C annealed powder, with, nonetheless, a less important reflectance in the NIR range. Interestingly, this shows that a higher W^5+^ concentration is maintained after a post-annealing treatment at 400 °C than compared to 600 °C. Furthermore, this observation can be related to the lower crystallinity of the annealed powder at 400 °C in comparison with the 600 °C annealed sample. This study thus shows that it is possible to modulate the absorption intensity in infrared by varying the crystallinity of the product through the control of the heat treatment temperature.

### 3.3. Effect of the Post Annealing Atmosphere at 600 °C

After synthesis, the particles were washed and dried in an oven (80 °C), then heat-treated at 600 °C under different atmospheres ranging from air (oxygen partial pressure equal to 0.2 bar) to argon (about 10^−5^ for the oxygen partial pressure) and oxygen partial pressures of 10^−2^, 10^−3^ and 10^−4^ bar obtained by mixing argon and air gas flows.

The crystalline structure of the powders was characterized by X-ray diffraction; the patterns obtained are shown in Figure 11. The powders annealed at 600 °C under argon and at oxygen partial pressures 10^−4^ bar exhibit a low degree of crystallization and their XRD patterns can be indexed in the Pm-3m cubic space group. Powders annealed at a p(O_2_) higher than 10^−3^ bar crystallized with the monoclinic structure. This set of data indicates that at a fixed annealing temperature, the annealing atmosphere greatly influences the crystal structure of the synthesized particles.

The products have then been characterized by EPR at low temperature (Figure 12). As previously shown, the signal at *g* = 2.0026, with a higher intensity for the argon annealed powder, than for the annealed powder at 10^−4^ bar, corresponds to the free electrons created within the materials by the reducing atmosphere. As soon as the atmosphere becomes more oxidizing, the signal assigned to free electrons disappears, i.e., for the powders annealed at 10^−3^ bar and 10^−2^ bar (Figure 12).

The optical properties of the powders obtained by diffuse reflection are reported in Figure 13. The compounds can be divided into two series: the ones annealed with the oxygen partial pressure equal to 10^−4^ or 10^−5^ bar (dark samples), and the others obtained after oxidative thermal treatment, i.e., for the oxygen partial pressure equal to 0.2 (air), 10^−2^, 10^−3^ bar, (clear samples). In the first series, the whole part of the spectrum, UV-visible-NIR range, is subjected to a strong and wide absorption. The annealed powder at 10^−4^ bar shows a behavior very close to the powder annealed under argon. Nonetheless, a low intense reflectivity windows emerges for the 10^−4^ bar sample; in fact, this material has a diffuse reflectance spectrum, which can be described as intermediate between the non-annealed powder (blue sample) and the argon annealed sample (black powder). In the second series, the comparison between the sample annealed at 10^−3^ bar, 10^−2^ bar and under air atmosphere, shows a progressive divergence in their reflectance spectra: the higher the oxygen partial pressure is, the lower intensity the inter-valence band responsible for the visible-near infrared absorption is (centered at about 2 micron wavelength). Consequently, the maximum of reflectivity percentage located at 500 nm increases from 76% for the 10^−3^ bar sample, to 81% for the 10^−2^ bar sample and up to 85% for the sample annealed under air. The luminosity L*, directly linked to this maximum of reflectance, represents the adequate parameter to quantify the phenomenon. The L* value increases for the three as-prepared sample from 87.96, to 91.00 and then 93.18, as the oxidative behavior of the annealing atmosphere increases. At 2000 nm, the comparison of the reflectivity percentage between the three samples of this second series exhibits the same behavior: with a value varying from 15% for the 10^−3^ bar sample, to 18% for the 10^−2^ bar sample and up to 40% for the sample annealed under air. Interestingly, the reflectance contrast (ΔR) between 500 and 2000 nm is only 45% for the sample obtained under air, whereas it reaches more than 60% for the two samples obtained at an oxygen partial pressure of 10^−2^ or 10^−3^ bar. Hence, these two latter samples, exhibit the best selectivity for the use of these compounds as solar filter, i.e., with the best contrast between the visible and near-infrared transmission. The optical properties of the two last samples, in an applicative point of view, would be very interesting for their use as passive heat regulators if they were shaped as thin films on glass windows.

The compounds obtained after a post-annealing treatment at 600 °C, but using variable oxygen partial pressures, has shown that the particles of WO_3_ are very sensitive to the synthesis atmosphere, i.e., their chemical composition (W/O ratio), their crystallite size, their crystallographic structure and their optical properties. Indeed, a very slight change in the oxygen concentration in the synthesis atmosphere drastically impacts the tungsten oxide physico-chemical properties. The sample with a monoclinic structure obtained after an annealing treatment at 10^−2^ bar is very clear colored, and has an average reflection of about 60% (average percentage of reflection in the visible range from 400 to 800 nm) in the visible range and 20% in the infrared range, whereas the sample obtained after an annealing treatment at 10^−4^ bar is very dark colored, with an average visible reflection of less than 5%, with a pseudo-cubic structure. In the next part, the photochromic behavior of the as-prepared samples is tested.

### 3.4. Photochromic Behavior of the Various as-Prepared Compounds

The last part of this study deals with the optical photochromic response of the as-prepared powders under UV irradiation. For this purpose, in the first part, the diffuse reflectance spectra on the whole UV-visible-NIR range of three reference samples (black, blue and yellow samples) were characterized before UV irradiation and after 90 min irradiation, using a UV source (mercury lamp with 254 nm wavelength). Ex-situ diffuse reflection spectra were also recorded versus irradiation duration on the raw materials (blue WO_3_). The results obtained are gathered in Figure 14a,b.

It can be observed that the photochromic behavior drastically depends on the starting sample. The yellow sample exhibits a very low photochromic behavior, i.e., the diffuse reflectance spectrum obtained after 90 min. irradiation being superimposable to the starting one. Hence, the monoclinic phase, with 50 nm crystallites, exhibits poor photochromism. This could be explained by a very rapid recombination of the exciton pairs (electron created in conduction band and holes created in valence band when the compound is UV irradiated). H_2_O has been proposed to be at the origin of the photochromism observed on WO_3_ thanks to a redox reaction between water and electron holes leading to tungsten ion photo-reduction [42,43]. In our condition this phenomenon is unlikely to occur and can be considered negligible. One can notice that the black sample does not exhibit photochromic behavior. For this second sample, the lack of measureable photochromism is more likely due to the occurrence of a very dark starting coloration assigned to a big amount of absorbing W^5+^ ions already present in the material than due to the absence of the redox process involving tungsten ions. Finally, on the opposite of the two previous samples, the blue sample exhibits large photochromism as compared with previous investigation reported on WO_3_ photochromism [7,44]. The DC optical contrast in the L*a*b* system between the sample non-irradiated and the sample irradiated for 90 min is about 25, which is more than four times the optical contrast that can be easily detected by human eye sensitivity. A kinetic study was then conducted on this product and the results are presented in Figure 14b, with ex-situ diffuse reflectance performed on samples after 15, 30, 60 and 90 min of irradiation. A fast response of the photochromism was evidenced, with coloration almost stabilized for irradiation times above 15 min.

The photochromism analysis was extended to two representative compounds elaborated at different temperatures and different atmospheres, for which a drastic change of the chemical, structural, morphological and optical properties was observed, i.e., the samples placed in both sides of the border delimiting the two samples series defined by the drastic change of physico-chemical properties: the first sample is the sub-micronic monoclinic compound with the more intense near infrared absorption (and lowest L* parameter, in the monoclinic series). The second sample is the pseudo-cubic NP with the more saturated blue coloration (lowest b*parameter in the pseudo-cubic series) (Figure 15).

The two samples exhibit a photochromic effect mainly concentrated in the near infrared part of the spectrum for the darkest monoclinic sample, characterized by a low visible DC contrast (about 11.6) but a DR (contrast in the percentage of reflection) at 1200 nm of about 30%. The photochromic behavior is only limited to the visible region (from the total absorption of the infrared part of the spectrum) for the bluest pseudo-cubic sample with a DC visible contrast of about 15. The photo-reduction of some W^6+^ ions at the origin of the photochromic behavior is then not exclusive to the pseudo-cubic structures. Nevertheless, synthesis conditions allowing the presence of some preexisting W^5+^ ions and oxygen vacancies are essential for promoting effective photochromic behavior.

Finally, in a last study, a focus is made on the reversibility of the photochromic effect (bleaching while the sample is protected from UV exposure); the study is performed on the blue sample. The results are reported in Figure 16.

The reversibility study shows that the photochromic effect is reversible in a large part; however, the kinetic of the reversibility is low in comparison with the kinetic of the coloring effect obtained versus irradiation time. Indeed, 5 h after irradiation, the reversibility reached only 40%. Hence the full recovery process can be apprehended to be about 15–20 h, which is quite long, besides large scale applications. Considering applications for such powder as its use in solar filter for smart windows, it will be shown that the bleaching phenomenon exhibits a complex behavior, with an impact on the bleaching kinetic of variable parameters as the time, the power and the wavelength of irradiation. Furthermore, current research on these materials have just shown that depending on the shaping process from the as-prepared powder samples to thin films: i.e., with or without the use of glass/polymer matrix addition, with the possibility to coat multilayered-systems, etc., the kinetic of the bleaching step is strongly impacted.

## 4. Conclusions

In conclusion, we developed a polyol synthesis process to obtain WO_3-x_ NPs with tenable chemical, morphological, structural, and optical properties. This low temperature chemical route process offers a large-scale production of tungsten oxide versatile nanostructures. The as-prepared WO_3-x_ NPs with oxygen sub-stoichiometric concentration, showed a very broad range of coloration from black to yellow (through blue-green colorations). Reducing conditions limit the crystallite growth, leading to NPs with pseudo-cubic structure with large oxygen sub-stoichiometry and with dark coloration, while oxidizing conditions promote the monoclinic structure organization with crystallites growth, with 50 nm diameter isotropic crystallites evidencing a light coloration. The monoclinic compounds obtained under the most reducing conditions exhibit an interesting reflectance contrast between UV, visible and NIR parts of the spectrum. Some of the as-prepared compounds exhibit a rapid and significant photochromic behavior. This photochromic behavior is highly dependent on the starting compound, showing that the photo-reduction of the tungsten ion is associated with the occurrence of pre-existing oxygen vacancies in the sample. A rapid and important change of colors was evidenced. The developed polyol chemical route clearly offers new perspectives of application for WO_3_ NPs as solar filter in smart windows, or illustration (the transparence window of these compounds being limited to the visible part of the spectrum) and also for UV and chemical sensing applications.

## Figures and Tables

**Figure 1 nanomaterials-09-01555-f001:**
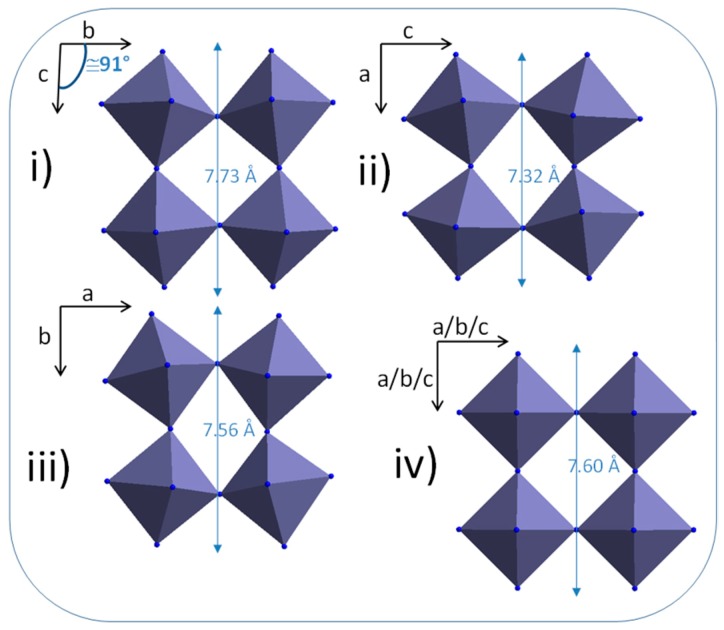
Representation of the octahedral site framework along a, b and c axis projection in the tungsten trioxide (WO_3_) monoclinic cell (**i**–**iii**,) and average pseudo-cubic cell equivalent representation (**iv**).

**Figure 2 nanomaterials-09-01555-f002:**
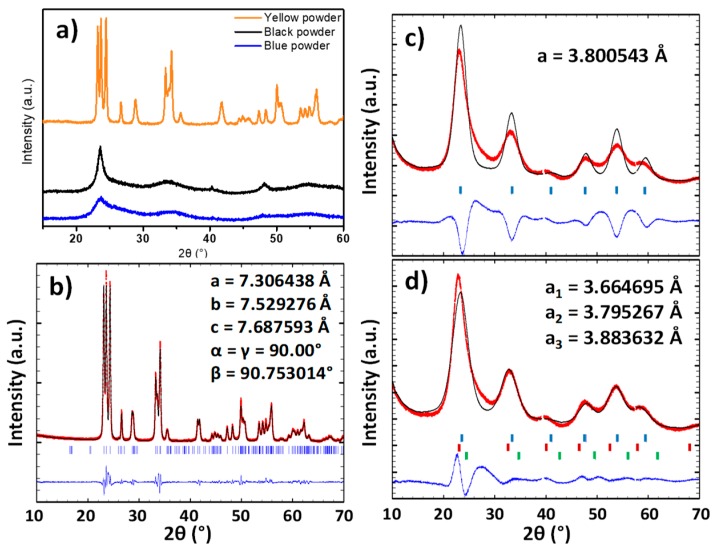
**a**) X-ray patterns of the blue, black and yellow samples; **b**) pattern matching refinement for the yellow sample considering a monoclinic phase; **c**) pattern matching refinement for the blue sample with one cubic unit-cell; **d**) pattern matching refinement for the blue sample with three superimposed cubic unit-cells.

**Figure 3 nanomaterials-09-01555-f003:**
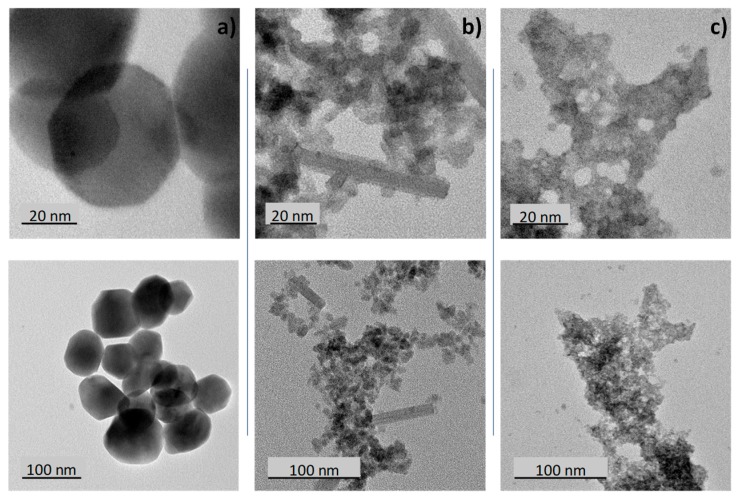
**Transmission Electronic Microscopy** (TEM) micrographs of WO_3_ powders: **a**) yellow sample, **b**) black sample, **c**) blue sample.

**Figure 4 nanomaterials-09-01555-f004:**
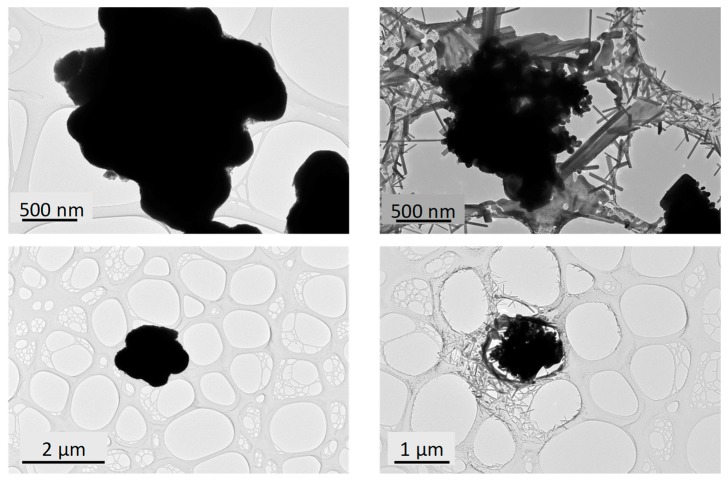
TEM images of WO_3_ black sample just at the beginning of the observation (left panel) and after a few minutes of electron irradiation (right panel).

**Figure 5 nanomaterials-09-01555-f005:**
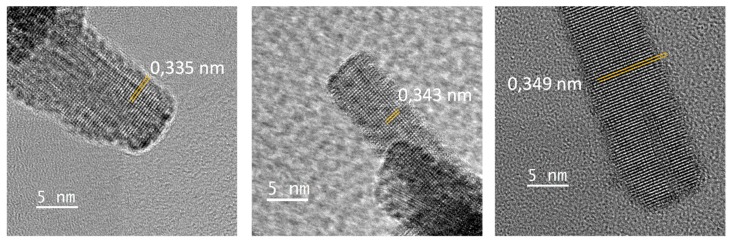
HR-TEM images of some created rod-like crystallite of the WO_3_ black sample after few minutes of electron irradiation.

**Figure 6 nanomaterials-09-01555-f006:**
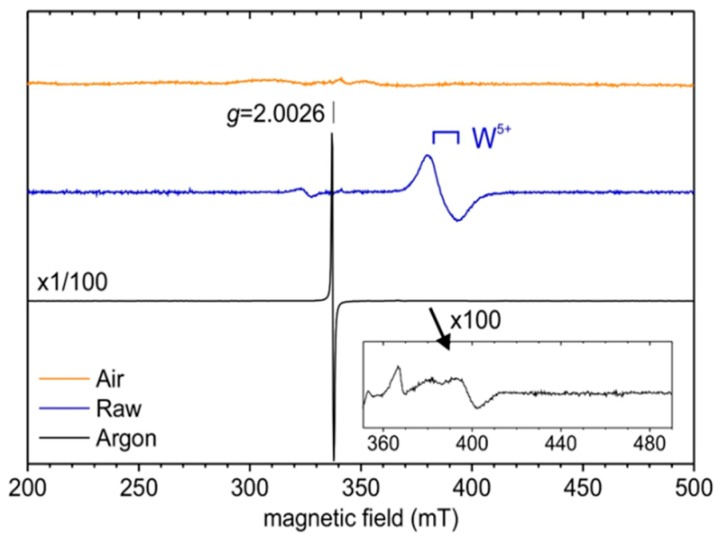
Low temperature (5 K) X-band EPR spectra of raw and annealed (under air or argon) WO_3_ powders.

**Figure 7 nanomaterials-09-01555-f007:**
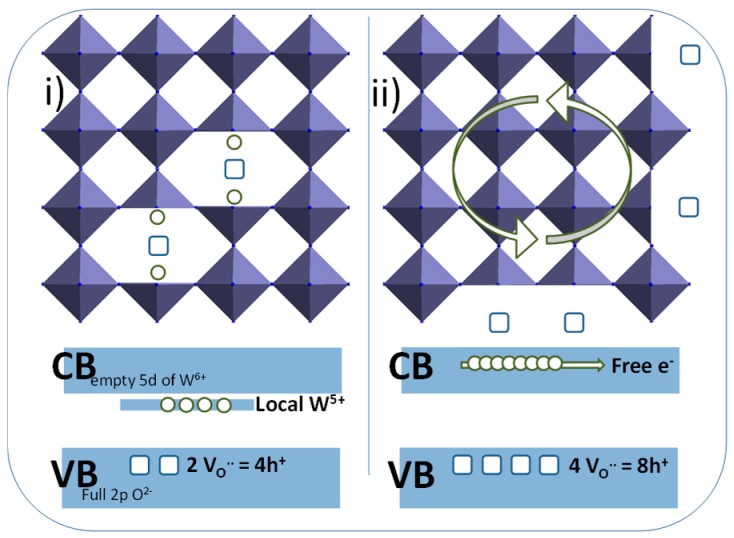
Schematic representation of the blue and the black samples, with simplified crystallographic representation and associated energy band diagrams.

**Figure 8 nanomaterials-09-01555-f008:**
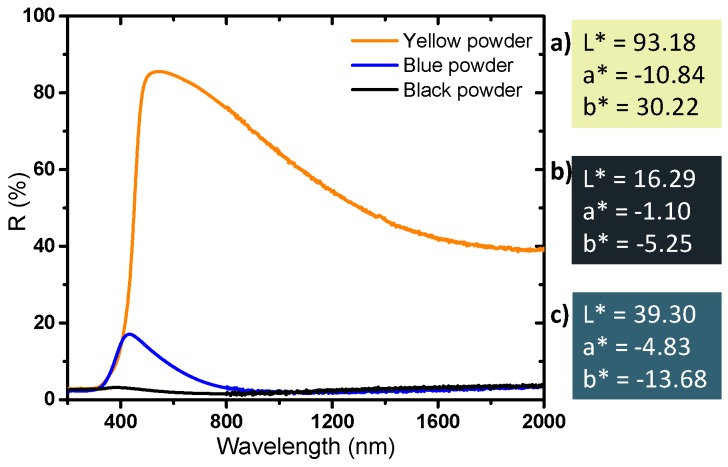
Diffuse reflectance of the powders and L*a*b* parameters coloration of the different WO_3_ compounds. **a**) Yellow, **b**) black, **c**) blue powder.

**Figure 9 nanomaterials-09-01555-f009:**
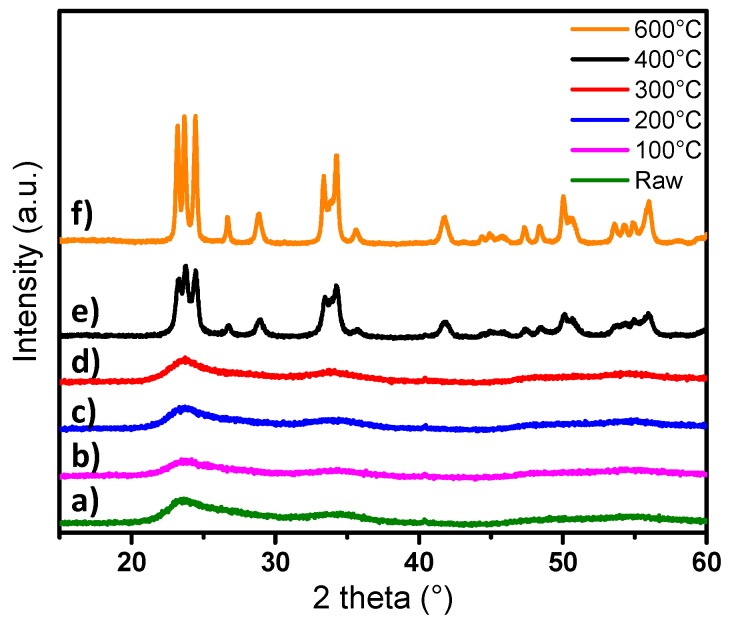
XRD patterns of WO_3_ synthesized by polyol process annealed under air at different temperatures: **a**) raw blue sample, **b**) 100, **c**) 200, **d**) 300, **e**) 400 and **f**) 600 °C.

**Figure 10 nanomaterials-09-01555-f010:**
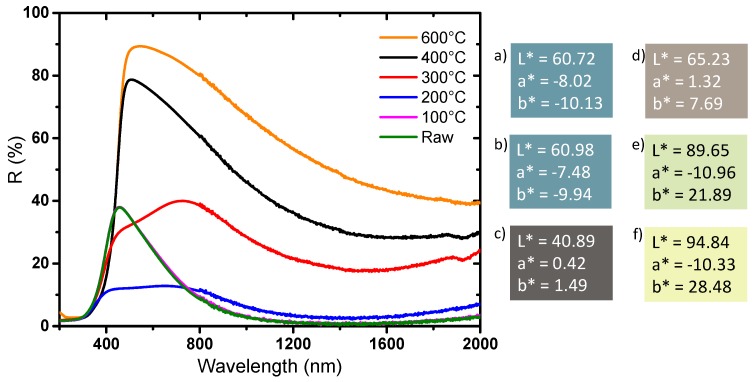
Diffuse reflectance and L*a*b* coloration parameters of the powders annealed under air at different temperatures: **a**) raw, **b**) 100, **c**) 200, **d**) 300, **e**) 400 and **f**) 600 °C.

**Figure 11 nanomaterials-09-01555-f011:**
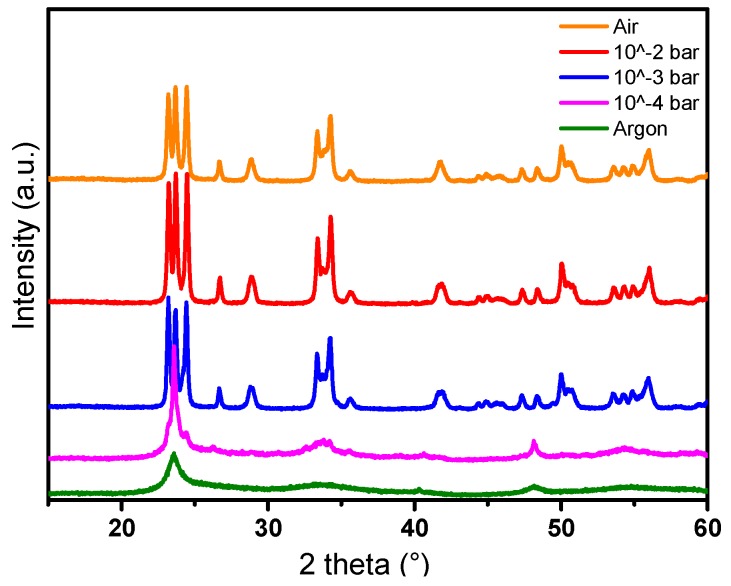
XRD patterns of WO_3_ powders synthesized by polyol process and annealed at 600 °C under different oxygen partial pressures: **a**) under air, **b**) p(O_2_) = 10^−2^ bar, **c**) t p(O_2_) = 10^−3^ bar, **d**) p(O_2_) = 10^−4^ bar, **e**) under argon.

**Figure 12 nanomaterials-09-01555-f012:**
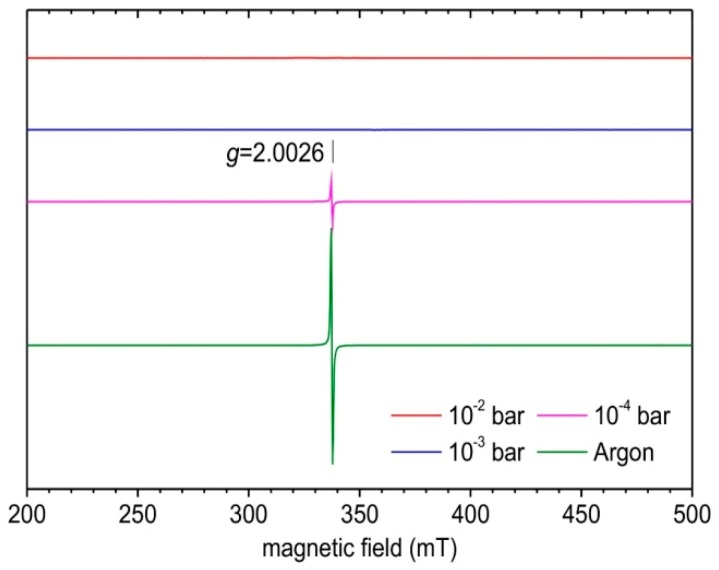
Low-temperature X-band EPR spectra of the samples obtained after different post-annealing treatment at 600 °C.

**Figure 13 nanomaterials-09-01555-f013:**
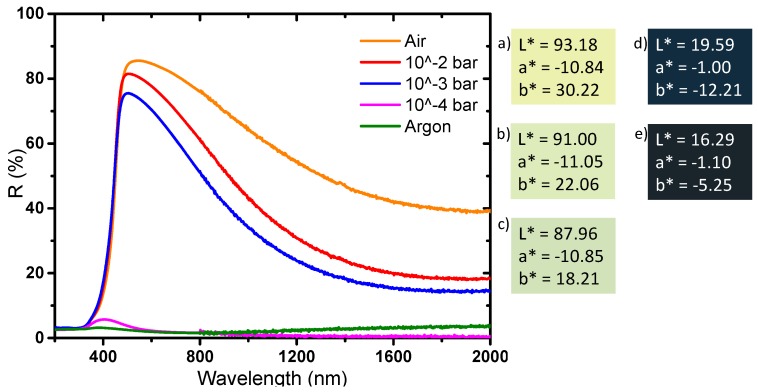
Diffuse reflectance and L*a*b* coloration parameters of the powders annealed at 600 °C under different oxygen partial pressures: **a**) air, **b**) 10^−2^ bar **c**) 10 ^−3^ bar **d**)10^−4^ bar and **e**) argon.

**Figure 14 nanomaterials-09-01555-f014:**
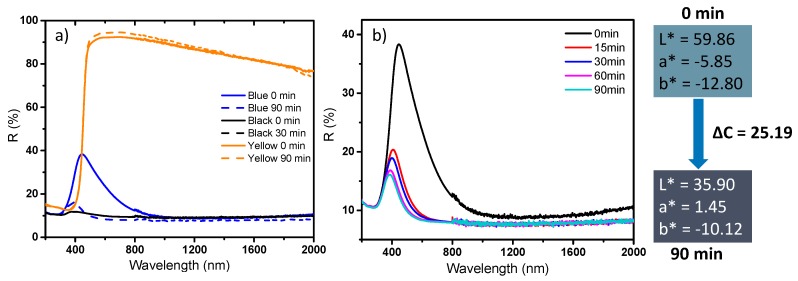
Photochromic behavior of the black, the blue and the yellow sample after different UV (254 nm) irradiation times. **a**) Comparison of the three samples before irradiation and after 90 min irradiation. **b**) Kinetic study of the photochromic behavior of the blue sample.

**Figure 15 nanomaterials-09-01555-f015:**
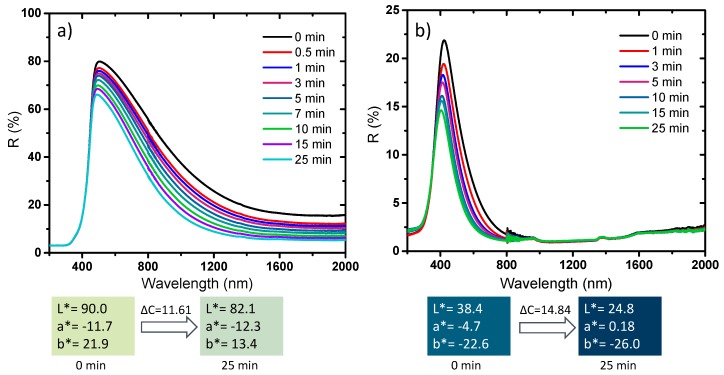
Photochromic behavior of the blue and the yellow samples after different UV (254 nm) irradiation times: **a**) the darkest monoclinic sample; **b**) the bluest pseudo-cubic sample.

**Figure 16 nanomaterials-09-01555-f016:**
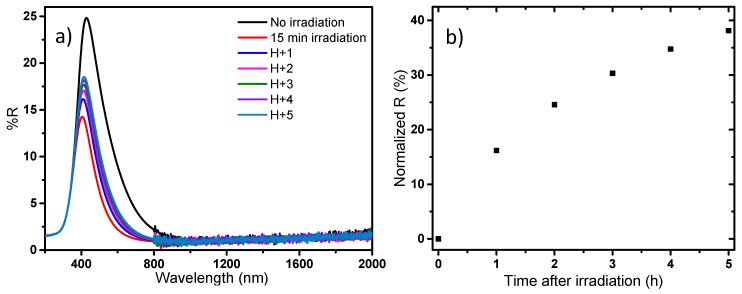
Reversibility of the photochromic behavior of the blue sample after 15 min (254 nm) irradiation time: **a**) reflectance spectra for different times after irradiation; **b**) evolution of the normalized reflectance at 500 nm versus time after irradiation.

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
