# Peer review of "Nanoparticles (NPs) of WO_3-x_ Compounds by Polyol Route with Enhanced Photochromic Properties"

_nanomaterials, 2019, doi:10.3390/nano9111555_

Round 1
Reviewer 1 Report
This study describes a polyolol synthesis method to obtain WO3-x nanoparticles with tuneable chemical and optical properties. The research was well designed, the methods and results are clearly described and discussed. Generally the paper is interesting and fits well to this journal.
Concerning the photochromic behavior of the samples, the recovery process (time of recovery and spectra for the different samples irradiated) is not presented and it should be to complete this characterization.
Author Response
Concerning the photochromic behavior of the samples, the recovery process (time of recovery and spectra for the different samples irradiated) is now presented at the end of the paper (with additionnal Figure: Figure 16 and associated comments) in order to complete the characterization.
Thank to the reviewer for the claim of this addition improving our the level od investigations.
Reviewer 2 Report
I though this was a good manuscript describing a careful and detailed study of tungsten oxide nanoparticles, certainly publishable with little revision. There were many spelling errors and a few glaring grammatical problems (examples: line 52 the first word should be "Regardless" not "Whatever", line 59 should read "may be difficult to discern"). Also, the scale bars on several TEM images were not legible, so I suggest changing to black line/black text.
Author Response
Dear reviewer,
The indicated grammar errors were corrected. A careful additionnal review of the text allowed detecting some additionnal typos.
Also, we modified the scale bars on the TEM micrographs with a clear background (Figures 3 & 4) to black line / black text (we keep white line /white text for the Figure 5, for which the image's background is dark grey.
Reviewer 3 Report
This is a nice work dealing with photochromic properties of tungsten nano-oxide, in which the main merit of the authors is the development of a synthesis route able to confer different coloration to their semples. The study is performed with complementary techniques, the manuscript is well organized and deserves publication
I have only one criticism on EPR results
The authors should provide more details on parameters used in their experiments, in particular modulation amplitude and microwave power must be chosen carefully to avoid line-shape distortions.
The sentence “narrow and intense signal at g=2.0026 associated with the presence of free electrons” is ambiguous and does not clarify what free electrons the authors refer. For instance, the g factor nearly 2 may be associated with unpaired electrons localized on paramagnetic centers. The same comment applies to the description of the figure 12.
Author Response
On the reviewer's criticism on EPR results, we have now provided, the needed details on parameters used in their experiments, in particular modulation amplitude and microwave power must.
New experimental :
"Electron paramagnetic resonance experiments were performed from room temperature down to T = 5 K in order to identify the W5+ ions and/or the occurrence of free electrons in the conduction band of the tungsten trioxide semi-conductors. The presented EPR spectra were recorded with a Bruker EMX spectrometer operating at X-band frequency (9.45 GHz) with 1 mW microwave power, 0.5 mT magnetic field modulation amplitude (frequency 100 kHz) and a spectral resolution of 0.15 mT/pt."
Also, The sentence “narrow and intense signal at g=2.0026 associated with the presence of free electrons” which was said as ambiguous was modified :
Hoping that the new version ion our new manuscript is now mode clear:
"This isotropic EPR signal with g-factor, g = 2.0026 ± 0.0002, close to the value of the free electron (ge = 2.0023) may be associated with conduction electrons rather than paramagnetic centres (such as electron trapped at oxygen vacancy) which usually exhibit anisotropic signals (axial or orthorhombic). At low temperature (5 K), additional weak orthorhombic signals are detected at higher magnetic fields (350-420 mT region). These last EPR signals characterized by g-values close to 1.7 may be due to W5+ (d1) ions in distorted environments, probably located close to the crystallite surface [35,36]. For the raw (blue) sample, the axial EPR signal with g^=1.76 and g//=1.71 is observed at low temperature (5 K) and can unambiguously be associated with W5+ ions in a more regular site (probably with C4v symmetry) within the WO3 structure [37–39]."
Reviewer 4 Report
I have just a minor comment...the abstract has to be reformulated and shortened to include the main results.
Author Response
The abstract was reformulated and shoterned in order to show only the main result, as claimed by the reviewer...
New abstract became :
Tungsten trioxide (WO3) is well-known as one of the most promising chromogenic compound, i.e. with a drastic change of coloration induced from different external stimuli and so applications are developed as gas sensors, electrochromic panels or photochromic sensors. This paper focuses on the photochromic properties of nano-WO3, with tunable composition (with tunable oxygen sub-stoichiometry). Three reference samples, with respectively yellow, blue and black colors, were prepared from polyol synthesis followed by respectively, post annealing under air, none post-annealing treatment, or a post-annealing under argon atmosphere. These three samples differ in terms of crystallographic structure (cubic system versus monoclinic system), oxygen vacancy concentration, electronic band diagram with occurrence of free or trapped electrons and their photochromic behavior. Constituting one main finding, it is shown that the photochromic behavior is highly dependent on the compound’s composition/color. Rapid and important change of coloration under UV irradiation was evidenced especially on the blue compound, i.e., the photochromic coloring efficiency of this compound in terms of contrast between bleached and colored phase as kinetic aspect is high. The photochromism is reversible in few hours. This opens hence a new window for the use of tungsten oxide as smart photochromic compounds.